# Sexual violence against migrants and asylum seekers. The experience of the MSF clinic on Lesvos Island, Greece

Rea A. Belanteri[1]*, Sven Gudmund Hinderaker[2], Ewan Wilkinson[3], Maria Episkopou[4], Collins Timire[5,6,7], Eva De Plecker[8], Mzwandile Mabhala[9], Kudakwashe C. Takarinda[5,6,7], Rafael Van den Bergh[10]

1 Médecins Sans Frontières—Operational Centre Brussels, Lesvos, Greece, 2 University of Bergen, Bergen, Norway, 3 Institute of Medicine, University of Chester, Chester, United Kingdom, 4 Médecins Sans Frontières-Operational Centre Brussels, Athens, Greece, 5 International Union Against Tuberculosis and Lung Disease, Harare, Zimbabwe, 6 International Union Against Tuberculosis and Lung Disease, Paris, France, 7 Ministry of Health and Child Care, AIDS and TB Department, Harare, Zimbabwe, 8 Médecins Sans Frontières-Operational Centre Brussels, Medical Department, Brussels, Belgium, 9 Department of Public Health and Well Being, University of Chester, Chester, United Kingdom, 10 Médecins Sans Frontières-Operational Centre Brussels, Operational Research Unit (LuxOR), Luxemburg, Luxemburg

* reabel24@gmail.com

**Data Availability Statement:** The authors confirm that, for approved reasons, some access restrictions apply to the data underlying the

## Abstract

### Objectives

Sexual violence can have a destructive impact on the lives of people. It is more common in unstable conditions such as during displacement or migration of people. On the Greek island of Lesvos, Médecins Sans Frontières provided medical care to survivors of sexual violence among the population of asylum seekers. This study describes the patterns of sexual violence reported by migrants and asylum seekers and the clinical care provided to them.

### Methods

This is a descriptive study, using routine program data. The study population consisted of migrants and asylum seekers treated for conditions related to sexual violence at the Médecins Sans Frontières clinic on Lesvos Island (September 2017-January 2018).

### Results

There were 215 survivors of sexual violence who presented for care, of whom 60 (28%) were male. The majority of incidents reported (94%) were cases of rape; 174 (81%) of survivors were from Africa and 185 (86%) of the incidents occurred over a month before presentation. Half the incidents (118) occurred in transit, mainly in Turkey, and 76 (35%) in the country of origin; 10 cases (5%) occurred on Lesvos. The perpetrator was known to the survivor in 23% of the cases. The need for mental health care exceeded the capacity of available mental care services.

findings. Due to the sensitive nature of sexual violence data, full datasets are not made available by default. Data are available through the MSF Data Sharing Agreement for researchers who meet the criteria for access to confidential data; requests should be addressed to the Data Sharing Agreement coordinator, Annick Antierens (Annick. Antierens@brussels.msf.org).

**Funding:** The programme was funded by the United Kingdom's Department for International Development (DFID); La Fondation Veuve Emile Metz-Tesch supported open access publications costs. The funders had no role in study design, data collection and analysis, decision to publish, or preparation of the manuscript.

**Competing interests:** The authors have declared that no competing interests exist.

## Conclusion

Even though the majority of cases delayed seeking medical care after the incident, it is crucial that access to mental health services is guaranteed for those in need. Such access and security measures for people in transit need to be put in place along migration routes, including in countries nominally considered safe, and secure routes need to be developed.

## Introduction

Sexual violence is widespread. It is reported to be particularly common in unstable environments such as where there are displaced populations and in conflict zones [1]. It is defined as '*any sexual act, attempt to obtain a sexual act, unwanted sexual comments or advances, or acts to traffic a person's sexuality, using coercion, threats of harm or physical force, by any person, regardless of relationship to the victim, in any setting, including but not limited to home and work*' [2]. Rape is an act of sexual violence and involves oral, anal or vaginal forced penetration by any part of the body or by any kind of object [2], and is undertaken against both sexes. Sexual violence can cause life-changing health and social problems for survivors [3].

There are many potential consequences of sexual violence, including sexually transmitted infections (STI), vaginal/rectal bleeding, other genital or body injuries, pain during sexual intercourse, unwanted pregnancy, psychosomatic issues, mental health problems, and even suicidal ideation, self-harm, and death [1, 4]. Longer term impacts may include behavioral problems, isolation, guilt, rejection by the family, or an inability to take care of the family, and may prevent a woman from marrying due to cultural taboos [5, 6]. People who have experienced sexual violence are also at higher risk of further attacks in the future [7]. Furthermore, it may also have a societal impact, and can considerably destabilize communities, in particular when used as a weapon of war [8].

Due to its stigmatizing nature, survivors of sexual violence do not always seek care, or may do so after a delay, limiting the medical care that can be provided (as treatments such as post-exposure prophylaxis for Human Immunodeficiency Virus (HIV) and other sexually transmitted diseases are contingent on early presentation). Sexual violence can affect both women and men [2], but men may find it more difficult to talk about it, as they are not expected to show weakness or lack of masculinity [9], or possibly because sexual violence services are linked to female health centers [10]. Consequently men may be more likely to delay or avoid seeking help for sexual violence.

A number of studies have focused on sexual violence in unstable contexts; of those, most were conducted in conflict or post-conflict zones [4, 11–13]. A more limited number of studies have focused on sexual violence among people in transit (this includes refugees, migrants, and asylum seekers); often in specific contexts, such as among migrants in transit in South and Central America [14–16] and among refugee populations during/following conflict [17, 18]. Out of all migration contexts, only a few studies have been published on sexual violence among displaced populations attempting to enter Europe. Keygnaert et al. [19] highlighted the risk of sexual violence that sub-Saharan migrants and asylum seekers faced when attempting to cross to Europe from Morocco. Freedman highlighted how current policies increased insecurity and vulnerability to sexual violence among women, and that insufficient medical and psychological support were being provided for female victims of violence in their countries of origin or in transit [20].

Sexual violence can and does occur anywhere along the route of displaced populations to safety. To be effective, medical care needs to be provided as soon as possible after the incident. Therefore a better understanding of the patterns of sexual violence and the needs of its

survivors in the context of migration is required, in order to improve and upscale the provision of care to this extraordinarily vulnerable population [21]. Health projects in migration settings offer an opportunity to understand and to document the risks and needs of migrants and asylum seekers who have experienced sexual violence. Ideally such projects will improve the screening process for sexual violence among people in transit and among people who are involuntarily contained for indeterminate periods of time, and result in better packages of care for sexual violence survivors.

The Greek island of Lesvos is very close to the Turkish border. It was the first arrival point and a geographically restricted site for a large number of people in transit seeking safety on European soil. The medical humanitarian organization Médecins Sans Frontières (MSF) operated a clinic for survivors of sexual violence on the island. The aim of this study was to describe the patterns of sexual violence experienced by migrants and asylum seekers, as reported to the MSF Clinic on Lesvos, and the clinical and mental health care provided to them, between September 2017 and January 2018.

## Materials and methods

### Design

This was a descriptive study using routine program data.

### General setting

Since early 2015, Greece has been at the forefront of the European refugee crisis. Greece is a country with many island communities: Lesvos is the third largest island of Greece, with an area of approximately 1,600 km$^2$. It is located in the northeast of the Aegean and is only 10 km from the east coast of Turkey. It is therefore the most accessibly island within the European Union. Its population is almost 86,500 [22]. In March 2016 the Balkan route for migrants travelling to the EU was closed and the EU-Turkey deal was implemented a few days later. According to the new regulations [23], asylum seekers had to complete their asylum procedure at the first landing point, the Greek islands, with no option to move to the mainland (hence a geographical restriction) unless they were deemed vulnerable. As a result, the islands became congested, accommodating many more migrants and asylum seekers than originally planned [24].

According to the United Nations High Commissioner for Refugees (UNHCR) official website, the population of Moria camp (the largest and most congested camp on Lesvos) reached 5,000 at the beginning of September 2017, overwhelming its original planned capacity of 2,300 [25]. Moreover, with only occasional transfers of people to the mainland and a total of 8,474 people arriving on Lesvos between September 2017 and January 2018, a vast increase in the population accommodated in Moria camp was seen over this period. By February 2018, the camp still had approximately 5,000 residents [25]. All camps on Lesvos were administered by the Greek authorities and UNHCR was present for the protection matters.

### Specific setting

MSF has provided health services on the island of Lesvos since July 2015. The Greek public health system was already overwhelmed by the economic crisis that started in 2009, with increased demand and decreased resources [26, 27]. Many of the people in transit on the islands had significant psychiatric needs, commonly due to trauma they faced in their country of origin or during the migration. Their needs exceeded the capacity of the trained staff in the local health care system, particularly the capacity to provide psychiatric care and trauma therapy. The lack of appropriately trained cultural mediators was also a problem.

As a response to these needs, MSF aimed to fill this gap by adapting its clinical services in the summer of 2017. A clinic was set up in Mytilene, the capital of Lesvos, for the provision of mental health care, medical care and social support for migrants and asylum seekers who had developed severe mental health disorders, including people who experienced torture (as defined in [28]) and sexual violence (as defined in [2]) at any point in their travels or prior to departure. A clinical database of sexual violence survivors was established when this clinic was set up.

In additional, an outpatient clinic was set up just outside Moria camp by the end of 2017. This was to provide paediatric, sexual and reproductive healthcare services. This including care for survivors of sexual violence presenting less than 120 hours after the assault (in contrast with the Mytilene clinic, where there was no such limitation on the timing of the incident). This time limit was established for the outpatient clinic because of the limited capacity of this clinic and taking into consideration the effectiveness of the medication offered in case of sexual violence. Data from all these sexual violence cases, of both locations, were entered in the same database.

Survivors of sexual violence with medical needs could either present themselves directly to one of the clinics requesting medical assistance, or could be referred to the sexual violence service by MSF staff from the other services (psychologist, doctor) or by another health Non-Governmental Organization (NGO). Mental health care was initially offered to migrants and asylum seekers who were either referred by MSF staff or who were self-referred. After September 2017, the mental health services also accepted referrals from other health NGOs, but no longer accepted self-referrals.

The medical and mental health care services were tailored to the needs of this group. Specifically, sexual violence survivors were offered a package of medical care based on the MSF protocol. This consisted of prevention of HIV (facilitating access to the hospital for Post Exposure Prophylaxis [PEP]), treatment of STI, emergency contraception, vaccination against Hepatitis B and Tetanus, and care of wounds or health complications after the sexual violence (Table 1). Mental health support as well as social support was available according to the patient's needs. Provision of a medical certificate was also offered, covering the medical examination of the patient and the incident as reported by the patient. Follow up appointments were scheduled based on the initial assessment of needs, with an average of three visits.

## Study population and period

The study included all male and female migrants and asylum seekers who sought clinical care for sexual violence at the MSF clinics on Lesvos Island, Greece between 1st September 2017 and 31st January 2018. This included individuals seeking care in the outpatient clinic in Moria, or in the Mytiline mental health clinic.

## Sources of data and variables

All the data were extracted from the standardized, pseudonymized MSF database for sexual violence (as described in [12]) and the waiting list of the mental health department of the MSF Lesvos Project. The outcome measurement was the number of survivors of sexual violence recorded. Other variables recorded were age, sex, nationality, location of incident (categorized as country of origin, in transit, on Lesvos), setting of incident (categorized as during migration activities [i.e. while on the move], during daily activities [i.e. any regular activity such as work or attending a market, not conducted at home], at home, in an institution [i.e. at school, church, prison], during an abduction situation, and others), time between incident and presenting for care (categorized as <72h, 72-120h, 5d-1m, 1m-1y, >1y), type of perpetrator, type

**Table 1. MSF protocol on health services offered at Lesvos MSF clinic to survivors of sexual violence, 2017–18.**

| Medical Management of Sexual Violence | |
|---|---|
| Interval § | Services recommended* |
| ≤72hours | Post exposure prophylaxis (PEP) for HIV** |
| | Emergency contraception*** |
| | STI prophylaxis or treatment † |
| | Tetanus vaccine |
| | Hepatitis B vaccine |
| | Mental health care |
| >72–120 hours | Emergency contraception |
| | STI prophylaxis or treatment |
| | Tetanus vaccine |
| | Hepatitis B vaccine |
| | Mental health care |
| >120 hours—6 months | STI prophylaxis or treatment |
| | Tetanus vaccine |
| | Hepatitis B vaccine |
| | Mental health care |

*All the services were provided after patient's consent and according to their health status and the type of sexual violence. Care of physical injuries and referral for termination of pregnancy (in case of a positive pregnancy test) was also offered. Additionally, medical certification was offered to all patients.

** In contrast with other contexts, in Greece MSF facilitates the access to PEP rather than administering it directly, due to the Greek legislation. PEP was generally offered only following penetration by the penis vaginally, anally or orally and only if presenting within 72 hours.

***Emergency contraception was offered in case of vaginal penetration, to every female ≥8 years old and/or after development of secondary characteristics of gender. Contraceptive methods offered were either pills or an Intrauterine Device.

§ Standardized time intervals as used in the MSF SV database

† Sexually Transmitted Infections

of sexual violence (categorized as rape, forced prostitution, and sexual touching), and associated violence. No information was collected systematically on the time spent in transit; anecdotal reports suggest this ranged from several weeks to several years, with 6 months being the median duration of travel.

## Analysis and statistics

The data analysis was performed using Epidata Analysis software version 2.2.2.186 (EpiData Association, Odense, Denmark). A descriptive analysis was done: means (standard deviations) were calculated for continuous data. Categorical data were summarised using frequencies and proportions. Groups were compared using the Chi-square test. P-values <0.05 were considered significant.

## Ethics approval

As *a posteriori* analysis of routinely collected programme data, the national ethics bodies in Greece did not consider this study as falling under their jurisdiction for ethics review. As the study was considered low risk and of public health importance, it was conducted under the exceptional approval of the medical director of Médecins Sans Frontières-Operational Centre Brussels.

## Results

Between September 2017 and January 2018, the MSF clinic on Lesvos recorded 215 patients presenting for care following sexual violence. The socio-demographic characteristics of these survivors and the location of the incident are shown in Table 2. Among these cases, 155 (72%)

**Table 2. Characteristics of 215 survivors of sexual violence, visiting the MSF clinic, Lesvos, Greece (September 2017-January 2018).**

| Characteristics | N | (%) |
|---|---|---|
| **Total** | 215 | (100) |
| **Age group (years)** | | |
| ≤10 | <5 | (<5) |
| 11–20 | 42 | (20) |
| 21–30 | 88 | (41) |
| 31–40 | 71 | (33) |
| >40 | 11 | (5) |
| Mean (SD) | 28.2 | (8.2) |
| **Sex** | | |
| Female | 155 | (72) |
| Male | 60 | (28) |
| **Current migrant camp** | | |
| Moria's camp | 208 | (96) |
| Karatepe's camp | 6 | (3) |
| Other | <5 | (5) |
| **Nationality groups** | | |
| Central and East Africa[1] | 95 | (44) |
| West Africa[2] | 78 | (36) |
| North Africa[3] | <5 | (<2) |
| Middle East[4] | 25 | (12) |
| Other | 12 | (6) |
| Not Recorded | 4 | (2) |
| **Location of incident** | | |
| Country of origin | 76 | (35) |
| In transit | | |
| *Turkey* | *106* | *(49)* |
| *Other transit country* | *12* | *(6)* |
| Lesvos | 10 | (5) |
| Other/unknown | 11 | (5) |
| **Interval between incident and presenting for care** | | |
| <72 h | 6 | (3) |
| 72–120 h | 0 | (0) |
| 5 days-1 month | 22 | (10) |
| 1 month-1 year | 174 | (81) |
| >1 year | 11 | (5) |
| Not recorded | 2 | (1) |

[1]Central and East: Congo, Democratic Republic of the Congo (DRC), Central African Republic (CAR), Ethiopia, Eritrea

[2]West: Cameroon, Nigeria, Mali, Burkina Faso, Gambia, Guinea (Conakry), Guinea Bissau

[3]North Africa: Morocco

[4]Middle East: Afghanistan, Iran, Iraq, Syria, Palestinian living in Syria

were female and 60 (28%) were male. The vast majority of the patients [208 (96%)] were living in Moria camp. The majority of the incidents of sexual violence [118 (55%)] occurred during migration, almost all of which took place in Turkey, and 76 (35%) occurred in the country of origin. Ten incidents (5%) in Lesvos were also documented. Only 6 (3%) of the cases presented in the appropriate timeframe to receive optimal care (<72 hours).

Characteristics of the cases seen by MSF, stratified by location of the event, are shown in Table 3. Delays in presentation were directly related to the attack occurring prior to arrival in Greece. A higher proportion of reported attacks on male survivors took place during migration period [40 (34%)], rather than in the country of origin [12(16%)]. The incident characteristics, stratified by location of the event, are presented in Table 4. Incidents tended to be more violent in the country of origin than in transit, with higher proportions of armed perpetrators (country of origin 45% vs transit 14%, p<0.01) and with more associated violence (country of origin 71% vs. transit 43%, p<0.01). Perpetrators of incidents during migration were more likely to be civilians who were unknown to the survivors.

**Table 3. Characteristics of 215 survivors of sexual violence presenting at the MSF clinic, stratified by location of the sexual violence incident, Lesvos, Greece (September 2017-January 2018).**

| Characteristics | Location of incident | | | | | |
|---|---|---|---|---|---|---|
| | **In Country of origin** | | **During Transit[5]** | | **On Lesvos** | |
| | n | (%) | n | (%) | n | (%) |
| **Total** | 76 | | 118 | | 10 | |
| **Age group (years)** | | | | | | |
| ≤10 | 1 | (1) | 1 | (1) | 0 | (0) |
| 11–20 | 14 | (18) | 21 | (18) | 4 | (40) |
| 21–30 | 33 | (43) | 51 | (43) | 2 | (20) |
| 31–40 | 22 | (29) | 41 | (35) | 3 | (30) |
| >40 | 6 | (8) | 4 | (3) | 1 | (10) |
| **Sex** | | | | | | |
| Female | 64 | (84) | 78 | (56) | 7 | (70) |
| Male | 12 | (16) | 40 | (34) | 3 | (30) |
| **Nationality groups** | | | | | | |
| Central and East Africa[1] | 38 | (50) | 47 | (40) | 6 | (60) |
| West Africa[2] | 22 | (29) | 49 | (42) | 1 | (10) |
| North Africa[3] | 0 | (0) | 1 | (1) | 0 | (0) |
| Middle East[4] | 11 | (14) | 12 | (10) | 2 | (20) |
| Other | 5 | (7) | 5 | (4) | 1 | (10) |
| Not Recorded | 0 | (0) | 4 | (3) | 0 | (0) |
| **Interval between incident and requesting care** | | | | | | |
| <72 h | 0 | (0) | 1 | (1) | 5 | (50) |
| 3 days-1 month | 1 | (1) | 20 | (17) | 1 | (10) |
| 1 month-1 year | 69 | (91) | 94 | (80) | 2 | (20) |
| >1 year | 6 | (8) | 3 | (3) | 1 | (10) |
| Not recorded | 0 | (0) | 0 | (0) | 1 | (10) |

1) Central and East Africa: Congo (Kinshasa) Congo (Brazzaville) Central African Republic, Ethiopia, Eritrea.

2) West Africa: Nigeria, Mali, Burkina Faso, Gambia, Guinea (Conakry) Guinea (Bissau) Cameroon.

3) North Africa: Morocco.

4) Middle East: Iran, Iraq, Syria, and Afghanistan.

5) Transit: any country where the victim resided at any point in time between leaving the country of origin and arriving in Lesvos.

**Table 4. Characteristics of 215 sexual violence incidents, stratified by location of the sexual violence incident, among survivors of sexual violence visiting the MSF clinic, Lesvos, Greece (September 2017-January 2018).**

| | Location of incident | | | | | |
|---|---|---|---|---|---|---|
| **Characteristics** | **In country of origin** | | **During Transit[1]** | | **On Lesvos** | |
| | N | (%) | N | (%) | N | (%) |
| **Total** | 76 | (100) | 118 | (100) | 10 | (100) |
| **Type of perpetrator** | | | | | | |
| Unknown civilian | 11 | (14) | 49 | (42) | 7 | (70) |
| Known civilian | 7 | (9) | 26 | (22) | 2 | (20) |
| Military | 17 | (22) | 1 | (1) | 0 | (0) |
| Smuggling groups | 3 | (4) | 13 | (11) | 0 | (0) |
| Family member | 11 | (14) | 3 | (3) | 0 | (0) |
| Institutional agent | 9 | (12) | 4 | (3) | 0 | (0) |
| Policeman | 6 | (8) | 1 | (1) | 0 | (0) |
| Organized gangs | 1 | (1) | 0 | (0) | 0 | (0) |
| Other | 2 | (3) | 4 | (3) | 0 | (0) |
| Not Recorded | 9 | (12) | 17 | (14) | 1 | (10) |
| **Armed Perpetrator** | | | | | | |
| No | 17 | (22) | 67 | (57) | 7 | (70) |
| Yes | 34 | (45) | 16 | (14) | 0 | (0) |
| Not recorded | 25 | (33) | 35 | (30) | 3 | (30) |
| **Type of event** | | | | | | |
| Rape | 74 | (97) | 107 | (91) | 10 | (100) |
| Sexual Slavery and forced prostitution | 1 | (1) | 8 | (7) | 0 | (0) |
| Sexual touching | 1 | (1) | 0 | (0) | 0 | (0) |
| Not Recorded | 0 | (0) | 3 | (3) | 0 | (0) |
| **Setting of incident** | | | | | | |
| During migration | 4 | (5) | 68 | (58) | 7 | (70) |
| Daily activity | 10 | (13) | 20 | (17) | 1 | (10) |
| Home | 27 | (36) | 3 | (3) | 0 | (0) |
| Institution | 14 | (18) | 7 | (6) | 1 | (10) |
| Abduction situation | 5 | (7) | 0 | (0) | 0 | (0) |
| Other | 1 | (1) | 1 | (1) | 0 | (0) |
| Not Recorded | 15 | (20) | 19 | (16) | 0 | (0) |
| **Associated violence:** | | | | | | |
| None | 22 | (30) | 67 | (57) | 7 | 70 |
| Beaten | 25 | (33) | 28 | (24) | 3 | (30) |
| Tortured | 9 | (12) | 4 | (3) | 0 | (0) |
| Witness of violence | 13 | (17) | 1 | (1) | 0 | (0) |
| Detained/ incarcerated | 4 | (5) | 4 | (3) | 0 | (0) |
| Robbed of property | 1 | (1) | 1 | (1) | 0 | (0) |
| Forced labor | 1 | (1) | 5 | (4) | 0 | (0) |

1) Transit: any country where the victim resided at any point in time between leaving the country of origin and arriving in Lesvos.

Health services provided by the MSF clinic to patients are shown in Table 5. Termination of pregnancy was requested by 3 out of the 10 pregnant women who attended the clinic. It could not be ascertained whether pregnancies were the result of the rape.

**Table 5. Health services provided to 215 survivors of sexual violence at the MSF clinic in Lesvos, Greece (September 2017—January 2018).**

| Health services | Females | | Males | |
|---|---|---|---|---|
| | Provided | | Provided | |
| | n | (%) | n | (%) |
| Total | 155 | | 60 | |
| Eligible for PEP[a] | 5 | (3) | 1 | (2) |
| Access to PEP (among eligible) | 5 | (100) | 1 | (100) |
| Eligible for Emergency Contraception | 5 | (3) | NA[b] | |
| Provision of Emergency Contraception (among eligible) | 3 | (60) | NA | |
| Sexually transmitted infections: prophylaxis/treatment | 92 | (59) | 29 | (48) |
| Tetanus vaccination | 66 | (43) | 26 | (43) |
| Hepatitis B vaccination | 64 | (41) | 26 | (43) |
| Physical examination [c] | 134 | (86) | 39 | (65) |
| Genital examination | 101 | (65) | 22 | (37) |
| Anal examination | 33 | (21) | 22 | (37) |

[a] PEP: post-exposure prophylaxis for HIV

[b] NA: not applicable

[c] Investigation for wounds

The total number of new referrals to mental health services at the MSF clinic (n = 825 in the whole period) and the number of clients on the waiting list by month are shown in Fig 1, where new referrals increased from less than 100 to more than 300 during November 2017, and then decreased to less than 100 again for the following month. The number of persons on the waiting list for mental health care increased from less than 100 to 553. The numbers increased substantially in November 2017, coinciding with referrals being accepted from other health NGOs, rather than only through MSF services and self-referrals. Due to the overloading, only very severe cases could be taken over by mental health department.

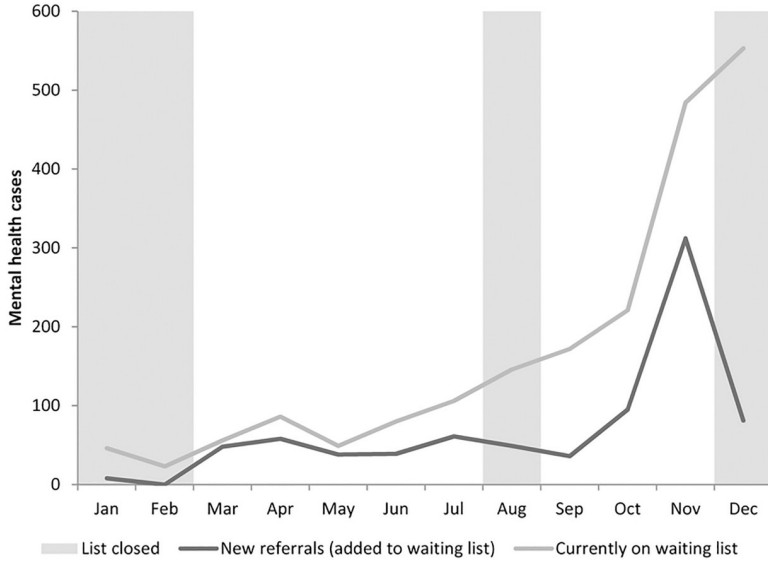

**Fig 1. New referrals to MSF mental health care and total number of patients on the mental health waiting list, Lesvos, 2017.**

## Discussion

This study on survivors of sexual violence among migrants and asylum seekers receiving care in the MSF clinic on Lesvos Island showed that almost all survivors had experienced the violence before reaching Lesvos. About a third of the incidents occurred in the country of origin and almost half during transit in Turkey. However, a number of incidents also occurred on Lesvos itself, reflecting a gap of the protection services on the island. Out of all types of sexual violence, rape accounted for more than 90% of the incidents reported.

A surprisingly high proportion of survivors (28%) were male, comparable with another survey by MSF which showed that 17% of all male refugees underwent sexual attacks in their efforts to leave from Central America [29]. Other facility-based studies in non-migration contexts present lower proportions of male survivors, including in an urban slum in Kenya (8%) and in post-conflict settings such as Liberia (2%) and eastern DRC (3%) [11–13]. However, our findings echo other studies in conflict areas of DRC, showing that 24% of adult males had experienced sexual violence at some point of their life [30], and in Lebanon with 20% of survivors being men and boys [31]. We do not know the reasons for the high rate of male survivors in our study. The fact that the clinic mainly offered mental health care services may have lowered the barrier for men to seek care, as sexual violence does not need to be disclosed at entry with such a setup. Also, the trust developed from patients towards the psychologist during mental health treatment might have led to disclosing and referral for medical care. Sexual violence to men may also be more prevalent in populations of migrants and asylum seekers, as it is a common component of torture. This is often directed at men and may be more frequent in such populations [32]. It may be used to humiliate and intimidate by officials and gang leaders who act as facilitators for the migration process (termed "migration professionals" in [19]). Our observation that sexual violence against males was more common during the migration phase supports this speculation.

While the majority of the asylum seeker population in Lesvos came from the Middle East, the most common countries of origin among sexual violence survivors were DRC and Cameroon, both for incidents perpetrated in the country of origin and for incidents occurring during transit. From our data, it is not possible to discern whether this reflects differences in prevalence of sexual violence according to the survivor's nationality, or differences in health-seeking behavior (possibly also linked to differing lengths of stay on Lesvos, depending on country of origin). Strikingly, our study showed that countries such as Turkey usually considered safe for migrants and asylum seekers, present a high risk of sexual violence; half of the reported sexual violence occurred there.

The perpetrator was known to the survivor (including family members) in 23%. This is a lower proportion than in more stable settings: in large urban and/or post-conflict settings, individuals known to the survivor represent 40–80% of the perpetrators [11–13, 33]. In conflict areas, perpetrators known to survivors tend to represent a much smaller proportion [12], and the same seems to hold true for migration contexts. This could be related either to underreporting of incidents when the perpetrators were known, or to the high exposure to unsafe situations during the migration period.

Many migrants and asylum seekers, including survivors of sexual violence, requested mental health support. The need for psychological care could not be met, reflected in the long waiting list after November 2017. Anecdotally, a volunteer doctor working in the camp reported finding the mental health services so overwhelmed there was no point in referring people who would normally benefit from mental health services.

We found that 94% of the survivors of sexual violence had been raped. This is much higher than what has been reported in other settings. In South America 60% of sexual violence

incidents concerned rape [34], whereas in the United States of America 21% of sexual violence survivors had been raped [35]. A suspicion of sexual slavery or trafficking was sometimes present during consultations in the clinic but there was no hard evidence, as the information revealed by the patients was not always complete or they had not understood what exactly had happened. As expected, sexual slavery/exploitation occurred more commonly during the highly vulnerable migration period. Other forms of sexual violence, such as intimate partner violence, may have remained underreported due to issues related to stigma, shame, and lack of protection services.

The medical care that could be offered to sexual violence survivors was usually relatively limited, as most sought care months after the event, limiting the treatment options available. The psychological impact of sexual violence and working in a cross-cultural manner was often complex. The lack of access to mental health care contributed to delays in the psychological assessment and adequate support of the patient. As psychological support is the mainstay of care for survivors presenting late, this placed a constraint on the quality of care provided. It was challenging for MSF to meet this huge need for mental health support.

Sexual violence services, including mental health support, should be an intrinsic component of care in all migration contexts, ideally offered along the route as well as in reception hotspots. Additionally, adequate security services need to be present to ensure that no new incidents of sexual violence can occur during the reception period, and to provide a sense of safety that will prevent deterioration of mental health conditions resulting from the history of sexual violence.

There is good evidence that countries considered safe such as Turkey (and Morocco, as in [19]) did not adequately protect people who were migrating through the country. This apparent lack of care is reinforced by the fact that Turkey, for example, offers only temporary protection to Syrian refugees, including permission to stay, basic rights and services [36].

It is recommended that health services should be available for survivors of sexual violence and to ensure people without documentation can still access protection services in these transit countries, as well as in EU Member States. The pursuit of externalization of migration policies is likely contributing to lack of safety of, and accountability to, displaced populations experiencing sexual violence.

We have shown that sexual violence care can be provided to this population, even providing access for male survivors, which is rarely achieved to this extent. The MSF model can thus be considered appropriate for identification and provision of care for both male and female survivors of violence in an asylum seeker/migrant population, though issues of scalability need to be examined, as shown by the long waiting times. Since MSF is an NGO, other actors, including governmental authorities, should take up similar programs in similar settings, particularly hotspots for new arrivals, to ensure prompt and comprehensive care for sexual violence survivors. The capacity of mental health care services must be properly planned to adequately meet the needs of this group.

This study had a number of limitations. There was a relatively small number of sexual violence survivors who attended the clinic, which limits the amount of data available for analysis of risk factors. Correct categorization of the cases was often difficult due to limitations imposed by the database and, in some cases, because of lack of full information disclosed by the patient. This was compounded by having to work cross-culturally and in different languages, which may have caused misunderstandings. All these may have resulted in missing or incorrect information. Additionally, it should be emphasized that this was a facility-based study, taking into consideration only those survivors who presented for care: there may thus be a selection bias against individuals who did not seek care in the first place, and/or an underrepresentation of sexual violence in general due to a reluctance to seek care.

## Conclusions

Sexual violence had been experienced by an appreciable number of migrants and asylum seekers arriving in Lesvos. The medical services that MSF provided were able to cope with the numbers, but the inadequate capacity of the mental health services made it difficult to refer and support all survivors of sexual violence who would have benefitted from such services.

When planning services to provide clinical care for survivors of sexual violence amongst migrants and asylum seekers, it is crucial that more attention is given to the mental health needs of this vulnerable group. Furthermore, provision of adequate clinical care and security measures for people in transit need to be put in place on routes that migrants use, including in countries, normally considered as safe for migrants.

## Acknowledgments

This research was conducted through the Structured Operational Research and Training Initiative (SORT IT), a global partnership led by the Special Programme for Research and Training in Tropical Diseases at the World Health Organization (WHO/TDR). The training model is based on a course developed jointly by the International Union Against Tuberculosis and Lung Disease (The Union) and Medécins Sans Frontières (MSF). The specific SORT IT program which resulted in this publication was implemented by: Medécins Sans Frontières, Brussels Operational Centre (OCB), Luxembourg and the Centre for Operational Research, The Union, Paris, France. Mentorship and the coordination/facilitation of these SORT IT workshops were provided through the Centre for Operational Research, The Union, Paris, France; the Operational Research Unit (LuxOR); AMPATH, Eldoret, Kenya; The Institute of Tropical Medicine, Antwerp, Belgium; The Centre for International Health, University of Bergen, Norway; University of Washington, USA; The Luxembourg Institute of Health, Luxembourg; The Institute of Medicine, University of Chester, UK; The National Institute for Medical Research, Muhimbili Medical Research Centre, Dar es Salaam, Tanzania.

We thank, also, Francisco De Bartolome Gisbert, medical officer of cell 2 MSF OCB, Sophie McCann, the advocacy manager in MSF Greece Mission and Declan Barry, medical coordinator of MSF OCB in Greece, for their contribution to our operational research.

## Author Contributions

**Conceptualization:** Rea A. Belanteri, Sven Gudmund Hinderaker, Ewan Wilkinson, Collins Timire, Eva De Plecker, Mzwamdile Mabhala, Kudakwashe C. Takarinda, Rafael Van den Bergh.

**Data curation:** Rea A. Belanteri.

**Formal analysis:** Rea A. Belanteri.

**Methodology:** Rea A. Belanteri, Sven Gudmund Hinderaker, Ewan Wilkinson, Collins Timire, Kudakwashe C. Takarinda, Rafael Van den Bergh.

**Project administration:** Maria Episkopou.

**Resources:** Rea A. Belanteri.

**Software:** Rea A. Belanteri, Collins Timire, Kudakwashe C. Takarinda.

**Supervision:** Rea A. Belanteri, Rafael Van den Bergh.

**Validation:** Rea A. Belanteri.

**Visualization:** Rea A. Belanteri.

**Writing – original draft:** Rea A. Belanteri.

**Writing – review & editing:** Rea A. Belanteri, Sven Gudmund Hinderaker, Ewan Wilkinson, Maria Episkopou, Collins Timire, Eva De Plecker, Mzwamdile Mabhala, Kudakwashe C. Takarinda, Rafael Van den Bergh.

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
