## [Decision Letter · Decision Letter 0]

19 Sep 2019

PONE-D-19-20179

Sexual violence against migrants and asylum seekers. The experience of the MSF clinic on Lesvos Island, Greece.

PLOS ONE

Dear Ms BELANTERI,

Thank you for submitting your manuscript to PLOS ONE. After careful consideration, we feel that it has merit but does not fully meet PLOS ONE’s publication criteria as it currently stands. Therefore, we invite you to submit a revised version of the manuscript that addresses the points raised during the review process.

We would appreciate receiving your revised manuscript by Nov 03 2019 11:59PM. To enhance the reproducibility of your results, we recommend that if applicable you deposit your laboratory protocols in protocols.io, where a protocol can be assigned its own identifier (DOI) such that it can be cited independently in the future. For instructions see: http://journals.plos.org/plosone/s/submission-guidelines#loc-laboratory-protocols

We look forward to receiving your revised manuscript.

Kind regards,

Lindsay Stark

Academic Editor

PLOS ONE

Journal Requirements:

2. You indicated that ethical approval was not necessary for your study. We understand that the framework for ethical oversight requirements for studies of this type may differ depending on the setting and we would appreciate some further clarification regarding your research. Could you please provide further details on why your study is exempt from the need for approval and confirmation from your institutional review board or research ethics committee (e.g., in the form of a letter or email correspondence) that ethics review was not necessary for this study? Please include a copy of the correspondence as an "Other" file.

Additional Editor Comments (if provided):

Reviewers' comments:

Reviewer's Responses to Questions

**Comments to the Author**

1. Is the manuscript technically sound, and do the data support the conclusions?

Reviewer #1: Partly

Reviewer #2: Partly

2. Has the statistical analysis been performed appropriately and rigorously? 

Reviewer #1: No

Reviewer #2: I Don't Know

3. Have the authors made all data underlying the findings in their manuscript fully available?

Reviewer #1: No

Reviewer #2: No

4. Is the manuscript presented in an intelligible fashion and written in standard English?

Reviewer #1: Yes

Reviewer #2: Yes

5. Review Comments to the Author

Reviewer #1: Very interesting topic, there is little known about forced migration to Greece, a country affected by a very interesting economic crisis. The authors have conveyed the magnitude of sexual violence and the devastating effect it has on displaced people. A valuable contribution of this manuscript is the information on male and female survivors of sexual abuse.

I have included my comments in an attached document

Reviewer #2: Thank you so much for this piece, which I was extremely excited to read. I found it very interesting, as it contains valuable data re: profiles of SV survivors and their perpetrators. I would be eager to have these insights from MSF’s records in the public domain and I think the article could be an important contribution to our understanding of sexual violence in migration, despite the small sample size.

I have a few specific notes and questions, then some global comments.

Specific notes and questions:

- It would be additionally valuable to identify any trends possible re: kinds of violence / perpetrators associated with different patient profiles – eg, what can we learn about who is at risk of what, where, from whom? As it is, the tables and discussion are largely static, with little analysis across data points. Data analysis would be strengthened by cross-sectional observations – eg, while noting that 23% of cases involved perpetrators known to victim, this could perhaps be disaggregated by national origin or kind of violence to better identify important situational trends and better understand disinclination to report to authorities, for example.

- Terminology: How were terms defined and phrased during survey? What is “compelled rape” as compared to "rape"? What is “daily activity”? Was torture clearly defined and understood? (Eg, “beaten” v “tortured” and also rape as a form of torture?). My suspicion is that the intake or screening instrument was a relatively blunt, which is not uncommon. However, this limitation or any reflections would be valuable to discuss, as conflation or confusion around terms of sexual violence seems to be a common challenge for all of us. Also, it is unclear whether forms of SV considered included intimate partner violence or sexual exploitation (latter of which was mentioned in conclusion and listed as an “associated violence” though unclear how defined or understood by MSF patients.) IPV can involve sexual violence and of course can have serious physical and psychological sequelae as well. Moreover, data indicate that rates of IPV remain high in conflict periods as well as in the context of forced displacement – is this major form of harm accounted for here? And if not, why not?

- I worry that lines 264-266 contain information that is not technically accurate. While it is true that medical evidence of vulnerability is helpful to an asylum-seeker ultimately, my understanding is that the vulnerability screening for migrants arriving in Greece is actually a pre-admissibility / procedural step that simply determines whether someone is exempt from the EU-Turkey deal, such that they would be permitted to apply for asylum in Greece. It is not technically part of the asylum application itself. This also matters because medical certification of vulnerability including sexual violence may be useful for protection purposes (eg, finding that one is exempted from Turkey bounce-back and can instead apply for asylum in Greece) without being legal relevant for a Convention-based claim for refugee protection (if the harm suffered or feared does not involve one’s country of origin, as seems to be the case with many migrants exploited and abused in transit.) I advise rewording after consultation with expert on Greek asylum process.

- I am also not sure about lines 267-272. There are some conclusory theories / statements re: why # of SV reports from Congolese v. Cameroonian migrants may differ – some statements re: exposure to conflict increasing risk of SV but also theories about how relatively long stays in camp may contribute to willingness to disclose SV. Is there data to support this theory? It may well be true but it seems there are insufficient data to infer these relationships. One could also theorize that different =people have different tendencies towards disclosure, either at group or individual level, not related to time spent in a camp.

- It may also be worth mentioning that medical certifications / records might be useful in proving torture cases or trafficking cases, not just asylum. (Though these forms of harm can also be the basis of an asylum claim.)

- Do the authors have any thoughts re: relative numbers (215 cases reported between Sept 2017 – Jan 2018)? Are these among new arrivals or general camp population? And if general population, why think so low compared to 5000 residents? It might help to explain how many of the total Moria resident population (for example) the MSF team actually reached, which might account for the winnowing down to 215 SV survivors. At any rate, this all raises the crucial issue of disclosure of sexual violence and how barriers may differ among groups and individuals – the discussion section touches on underreporting and disclosure but does not fully develop theories on it that are grounded in the data. This may be difficult with the limited data available but if there is any qualitative material in MSF records that could shed light on decisionmaking re disclosure, that would be fascinating indeed.

Global comments:

I think the data presented are incredibly valuable and these findings should be shared. However, as noted above, it would help to have clarification of MSF’s work on Lesvos and how the intake questionnaire was administered and translated, so we can better gauge how well the terms were understood by the migrants themselves.

Policy and practice recommendations seem extremely important, particularly re: a) provision of mental health support services, b) inclusion of male survivors in screening & support efforts, and c) false reliance on “safe countries” like Turkey and Morocco. To strengthen these points, more background info / explanation would be helpful.

- Eg, re: the mental health impacts of sexual and gender-based violence, as well as whether any mental health assessment was done with this population as part of MSF activities. The recommendation is sound but comes from out of nowhere.

- Eg, whether one sees higher rates reported among specific sub populations of patients (as in Afghan teens on the move, who received a degree of attention several years ago), which might indicate which men and boys are most vulnerable in what situations.

- Eg, more context re: EU-Turkey deal and its bottlenecking impact on Greece, including the assumption that Turkey is a safe country for return and adjudication. It could help, for example, to explain the major migration routes – as well as the short distance between Lesvos and Turkey, which would explain how the vast majority of migrants on Lesvos came through Turkey. Cites to reported human rights abuses against migrants in Turkey would help make the point re: false reliance on safe third country policy.

I would definitely welcome the eventual publication of these data and insights, as I think the data is so important and MSF’s crucial work on Lesvos warrants significant attention. However, to strengthen the piece, I do suggest a strengthening of the discussion section and close proofread and technical / stylistic edit by a native English speaker.

6. PLOS authors have the option to publish the peer review history of their article (what does this mean?). If published, this will include your full peer review and any attached files.

Reviewer #1: No

Reviewer #2: Yes: Kim Thuy Seelinger

---

## [Author Response · Author response to Decision Letter 0]

20 Feb 2020

To the editor, PLOS ONE

Dear editor,

Thank you for your message including the reviews of our paper. We have amended the paper following the comments, and we think the paper is better now. In our responses below we refer to line numbers in the new revised version with track changes. Reviewers’ comments are shown, and our response is given point by point in BOLD.

Response to Reviewers

2. You indicated that ethical approval was not necessary for your study. We understand that the framework for ethical oversight requirements for studies of this type may differ depending on the setting and we would appreciate some further clarification regarding your research. Could you please provide further details on why your study is exempt from the need for approval and confirmation from your institutional review board or research ethics committee (e.g., in the form of a letter or email correspondence) that ethics review was not necessary for this study? Please include a copy of the correspondence as an "Other" file.

 Thank you for this clarification question – the ethics situation around this study is indeed complex. We have submitted the study protocol to a number of national ethics bodies (including the Committee of Bioethics of the Medical School of the National and Kapodistrian University of Athens; the ethics committee of the University of Aegean; the ethics committee of the National School of Public Health; and the National Bioethics Committee). All the ethics committees we approached stated that the protocol fell outside of their jurisdiction: we include one such correspondence as “Other” file in the submission; we can provide an English translation if required. This reflects the general administrative gap that exists in Greece for ethics reviews for researchers who are not formally connected to an academic institution and who are not conducting clinical trials, as retrospective reviews of programme data are not covered in the national guidelines for ethics review. 

MSF has its own process for ethics review for studies that are based on routine programme data, which is granted if a set of pre-defined criteria are satisfied (in terms of data protection, general public health benefit, etc.). However, one of these criteria requires the “approval by the relevant national ethics bodies”. This could not be obtained in Greece for the reasons mentioned above. We have therefore obtained the specific, exceptional approval to conduct this study from the medical director of MSF-Operational Centre Brussels, considering the study’s low risk as it used routinely collected, anonymised programme data, retrospectively, and its public health importance. We have rephrased the ethics section accordingly.

Comments to the Author

Reviewer #1: 

General comments:

-Very interesting topic, there is little known about forced migration to Greece, a country affected by a very interesting economic crisis. The authors have conveyed the magnitude of sexual violence and the devastating effect it has on displaced people. A valuable contribution of this manuscript is the information on male and female survivors of sexual abuse.

We thank the reviewer for these supportive comments. 

An important concept that needs to be clarified are the categories of the location of the sexual assault. The three categories are country of origin, in transit or in Lesvos. It is unclear how long someone may be ‘in transit’ for. 

Thank you for raising this point. The time “in transit” can vary from weeks/months up to several years – this information is not collected routinely (to avoid patients feeling interrogated about their migration history). Anecdotal evidence suggests the median duration in transit was 6 months, and we have added this as estimate in the manuscript (Sources of data paragraph, as we do not wish to imply this was a study finding).

Timing is an important part of this analysis that needs clarification. There is a variable called ‘interval incidence-care’ that ranges from less than 72h to over a year. It seems that the clinic only sees acute cases (less than 72h), however 91% of cases presented within 1 month to 1 year since the incident. 

We have provided more detail of the time between the incident and presenting for care in table 2 and 3. Additionally, we have clarified in the setting section that there was indeed a time restriction of <120 hours in the outpatient clinic located directly outside the Moria camp, but that there were no time restrictions for patients accessing the more mental health-focused Mytiline clinic. 

This technical level of information needs to be considered in the methods section and in the analytic plan. Of importance, each variable should be carefully defined in the methods section. For example, there is discussion of PEP eligibility but not what the eligibility criteria is.

Thank you, we have attempted to provide all relevant definitions in table 1.

The final section on mental health services is interesting, however the manuscript is already filled with 5 tables on health services and detailed information on sexual violence. I would recommend removing the section on mental health or more clearly including it in the objective statement and analysis plan. It seems to me that the questions about volume of new referrals to the MSF clinic is different from the main line of inquiry related to sexual violence. 

As bridging the mental health care gap in contexts such as Lesvos is a major public health issue, we have opted to keep the MH service analysis in the manuscript, and have thus modified the objectives to include mental health care, see line 162 and 187-196 of the tracked version of the manuscript.

Currently, the manuscripts reads more like an organizational report and less like an academic article. Below is a list of recommendations intended to improve the scientific contribution of the manuscript and its readability to a wider audience. 

General comments:

-Kindly use the term ‘survivor’ in place of ‘victim’

-The manuscript should be reviewed for typo’s

-Rephrase sentences to start with words and not numbers

Tables

-suppress cell sizes under 5 to protect confidentiality

-review for consistency in font size, decimals 

-Review table titles for consistency

-Include the total n for each column in Tables 3 and 4

Thank you. We have made these alterations, with the exception of some of the smaller cell sizes, which were considered not to hold any risk for participant confidentiality.

Abstract

-Please present the frequency (n) and percentages for each variable

-The conclusion focuses on accessing services and mental health care, however there are not estimates on mental health presented in the results

-In place of words in quotations (e.g. “safe”), state the issue more objectively

We have modified the abstract according to these suggestions, and have expanded on the mental health aspect in the Results section. 

Introduction

-I would generally review the manuscript to ensure that the title, abstract and manuscript are all conveying the same message.

Thank you – we have gone through the full manuscript and have revised accordingly.

-The first three paragraphs of the introduction focus broadly on sexual violence and it’s consequences. While interesting, the most important contribution of this article is the problem of sexual violence in humanitarian settings, in which there is an extensive amount of literature that should be added to the introduction. The specific situation of Greece and migration through the Mediterranean Sea, Balkans or the surrounding geographic area is a critical area to include. The issues surrounding migration by water or to islands would add value to the introduction section and the literature in general. 

We have revised the introduction in accordance with the reviewer’s suggestions.

-In the fourth paragraph, the distinction between people in ‘transit or containment’ compared to those in conflict or post-conflict is unclear. The term containment is often used in infectious disease and outbreaks scenarios. Please clarify the term as it is used in this article or consider replacing it. 

Thank you. We have replaced the term containment.

-Please review the objective statement so that it clearly prepares the reader for the methods and results section. A lot of the methods and part of the results feature issues of mental health, however this is not covered in the introduction or the objectives. This should be reconciled.

Thank you. In the objectives we added “mental health”, and we give numbers of those seeking mental health care in results, see line 254-261 of the tracked version of the manuscript.

Methods

-There are six paragraphs dedicated to describing the setting. Please pick out the key points that summarize the setting and how it specifically relates to the research objectives and study design. I recommend moving some of the ‘General setting’ section to the introduction and include academic references

Thank you. We understand the point raised by the reviewer but we feel the introduction is already long and sets the general scene of sexual violence among populations in transit; we are concerned that moving parts of the setting to the introduction could confuse the reader, and could feed into the notion that this study is rather an organizational report, as highlighted by the reviewer above. We would prefer to leave the setting description as is, but if you feel strongly it should be moved we will do so. 

-In the Study population and period section, please describe the inclusion criteria and exclusion criteria. Is the sample restricted to people in the Moria camp who accessed the clinic and self-disclosed as survivors of sexual violence? Is the Moria camp an UNHCR camp? These details will clarify the external generalizability of the results.

The study population consists of all individuals seeking care for sexual violence with MSF – this includes people who sought care at the outpatient clinic (for incidents within the past 120 hours) and people who sought care in the Mytiline clinic. We have tried to clarify this further in the manuscript. Moria is a camp managed by the Greek authorities, but with a UNHCR presence. 

-All variables presented in the tables should be previously defined in the methods

Thank you. We have defined all the variables in lines 185-194 in the tracked version of the manuscript.

-Some of the statistics described in the methods section are not presented in the results section or tables. For example, the statistical analysis section describes using chi-square tests, however p-values, significance levels or chi-square estimates are not presented in the tables or text. There is mention of ‘groups’ but the group comparisons are not defined. Perhaps in transit vs Lesvos camp? Please clarify. If this is one of the main objectives of the analysis, it should be clearly stated in the objective statement and the analysis plan should be able to address the objectives

Thank you for highlighting this inconsistency – we performed statistical tests to compare incident typologies between transit and country of origin, which is now clearly stated in the results section.

-There are many cells with small cell sizes where chi-square tests are not the most appropriate statistical test. Small cell sizes below 5 should be suppressed to protect confidentiality.

Thank you. We have made these alterations, with the exception of some of the smaller cell sizes, which were considered not to hold any risk for participant confidentiality.

Results

-Please include n(%) in the results section

The results have been updated accordingly.

-Use numeric estimates in place of descriptors such as ‘a higher proportion’ or ‘increased substantially’. This is particularly important if significance testing was not performed. Keep the results section to describe the statistics and put the interpretations into the discussion section

We have attempted to reduce the use of qualifiers in the results section, and have added the proportions to the text. 

In the ‘Specific setting’ section it says that the Moria camp provides services for those who presented for care less than 120 hours following the assault. However, in the results it says that 81% experienced sexual violence between 1-12 months before getting care. Please review.

Thank you for highlighting this point of confusion – we have added clarification in the setting and study population paragraphs, to clarify that patients could present through two routes: the Moria outpatient clinic (seeing only urgent cases <120 hours) and the Mytiline mental health clinic where older cases of sexual violence were also seen. 

Table 4 as described in the text says ‘Incidents tended to be more violent in the country of origin, with higher proportions of armed perpetrators and with more associated violence’ - This has not be statistically evaluated

For this number we have added the p-value, line 228-231 in the tracked manuscript.

Figure 1 is unclear

⁃ ‘List Closed’ is an unclear category

 Thank you. We have amended Figure 1, and “list closed” has been replaced with “Intakes to waiting list suspended”.

Discussion

-Please clarify the issue with unofficial ‘migration professionals’ in paragraph three of the discussion

This term is used to refer to individuals who act as facilitators for migration in an official or unofficial capacity – now clarified in the text.

-Justify conclusions based on the shown results, avoid anecdotal information

We have tried to provide more clear justifications of the conclusions in the discussion section.

-In the discussion of known perpetrators, please related the finding to comparable populations. The study of police reported sexual assault cases in the UK is not an appropriate reference.

We agree that the setting is not similar, and we have provided a number of references from MSF settings instead. 

Limitations section

-Include description of the measurement bias, selection bias

We have added these in lines 356-359 in the tracked manuscript.

References

-Review for typos and inconsistent formatting

Thank you, corrections have been made.

Reviewer #2: 

Specific notes and questions:

- It would be additionally valuable to identify any trends possible re: kinds of violence / perpetrators associated with different patient profiles – eg, what can we learn about who is at risk of what, where, from whom? As it is, the tables and discussion are largely static, with little analysis across data points. Data analysis would be strengthened by cross-sectional observations – eg, while noting that 23% of cases involved perpetrators known to victim, this could perhaps be disaggregated by national origin or kind of violence to better identify important situational trends and better understand disinclination to report to authorities, for example. 

We agree that analysis of disaggregated data would be interesting with bigger datasets for identifying risk factors, but our dataset is insufficient for this. We would like to focus on the story as descriptive. Further disaggregation would, also, hold a risk for identification of individuals, and patient safety is our first concern.

- Terminology: How were terms defined and phrased during survey? What is “compelled rape” as compared to "rape"? What is “daily activity”? Was torture clearly defined and understood? (Eg, “beaten” v “tortured” and also rape as a form of torture?). My suspicion is that the intake or screening instrument was a relatively blunt, which is not uncommon. However, this limitation or any reflections would be valuable to discuss, as conflation or confusion around terms of sexual violence seems to be a common challenge for all of us. Also, it is unclear whether forms of SV considered included intimate partner violence or sexual exploitation (latter of which was mentioned in conclusion and listed as an “associated violence” though unclear how defined or understood by MSF patients.) IPV can involve sexual violence and of course can have serious physical and psychological sequelae as well. Moreover, data indicate that rates of IPV remain high in conflict periods as well as in the context of forced displacement – is this major form of harm accounted for here? And if not, why not?

The reviewer is correct to point out that the intake instrument was relatively blunt, and principally recorded what was reported by the survivor. However, a level of standardisation was maintained: the clinic accepts all cases of sexual violence according to the WHO definition (provided in the introduction), which can indeed include IPV and/or sexual exploitation. However, we believe this to be a minority among our patients, likely due to reporting bias (as stated more explicitly now in the limitations section): we anticipate that in an environment of high population density and exceedingly limited protection services, IPV will be considerably underreported (which is also seen in other MSF contexts). It is likely that a number of cases of rape by a “family member” are in fact cases of IPV. As these considerations are largely anecdotical, we opt not to develop them in detail in the manuscript. Concerning torture, MSF uses the ICRC definition, and due to the specialisation of the Mytiline clinic in providing care for victims of torture and ill-treatment, we believe this term to have been used appropriately by all staff. We have now also included the reference to the ICRC definition of torture in the manuscript, and in general have attempted to provide more clear definitions for all terms used.

- I worry that lines 264-266 contain information that is not technically accurate. While it is true that medical evidence of vulnerability is helpful to an asylum-seeker ultimately, my understanding is that the vulnerability screening for migrants arriving in Greece is actually a pre-admissibility / procedural step that simply determines whether someone is exempt from the EU-Turkey deal, such that they would be permitted to apply for asylum in Greece. It is not technically part of the asylum application itself. This also matters because medical certification of vulnerability including sexual violence may be useful for protection purposes (eg, finding that one is exempted from Turkey bounce-back and can instead apply for asylum in Greece) without being legal relevant for a Convention-based claim for refugee protection (if the harm suffered or feared does not involve one’s country of origin, as seems to be the case with many migrants exploited and abused in transit.) I advise rewording after consultation with expert on Greek asylum process. 

Thank you. We agree, and have opted to remove the paragraph. 

- I am also not sure about lines 267-272. There are some conclusory theories / statements re: why # of SV reports from Congolese v. Cameroonian migrants may differ – some statements re: exposure to conflict increasing risk of SV but also theories about how relatively long stays in camp may contribute to willingness to disclose SV. Is there data to support this theory? It may well be true but it seems there are insufficient data to infer these relationships. One could also theorize that different =people have different tendencies towards disclosure, either at group or individual level, not related to time spent in a camp. 

We agree that there may be difference in willingness to disclose and report according to the culture of the survivor, and we acknowledge that we do not have data on this. We have adapted the statements in discussion and limitations of the research.

- It may also be worth mentioning that medical certifications / records might be useful in proving torture cases or trafficking cases, not just asylum. (Though these forms of harm can also be the basis of an asylum claim.) 

Thank you for your comment. While we agree that this is an additional possible impact of certification, following the removal of the paragraph on medical evidence and vulnerability, we feel it may be confusing to introduce this information.

- Do the authors have any thoughts re: relative numbers (215 cases reported between Sept 2017 – Jan 2018)? Are these among new arrivals or general camp population? And if general population, why think so low compared to 5000 residents? It might help to explain how many of the total Moria resident population (for example) the MSF team actually reached, which might account for the winnowing down to 215 SV survivors. At any rate, this all raises the crucial issue of disclosure of sexual violence and how barriers may differ among groups and individuals – the discussion section touches on underreporting and disclosure but does not fully develop theories on it that are grounded in the data. This may be difficult with the limited data available but if there is any qualitative material in MSF records that could shed light on decision making re disclosure, that would be fascinating indeed.

We fully agree – unfortunately, there is no qualitative data available on this point. We have mentioned it more explicitly now as a limitation in discussion.

Global comments:

I think the data presented are incredibly valuable and these findings should be shared. However, as noted above, it would help to have clarification of MSF’s work on Lesvos and how the intake questionnaire was administered and translated, so we can better gauge how well the terms were understood by the migrants themselves.

Policy and practice recommendations seem extremely important, particularly re: a) provision of mental health support services, b) inclusion of male survivors in screening & support efforts, and c) false reliance on “safe countries” like Turkey and Morocco. To strengthen these points, more background info / explanation would be helpful.

- Eg, re: the mental health impacts of sexual and gender-based violence, as well as whether any mental health assessment was done with this population as part of MSF activities. The recommendation is sound but comes from out of nowhere.

- Eg, whether one sees higher rates reported among specific sub populations of patients (as in Afghan teens on the move, who received a degree of attention several years ago), which might indicate which men and boys are most vulnerable in what situations.

- Eg, more context re: EU-Turkey deal and its bottlenecking impact on Greece, including the assumption that Turkey is a safe country for return and adjudication. It could help, for example, to explain the major migration routes – as well as the short distance between Lesvos and Turkey, which would explain how the vast majority of migrants on Lesvos came through Turkey. Cites to reported human rights abuses against migrants in Turkey would help make the point re: false reliance on safe third country policy.

Thank you - see changes in lines: 109-113 of the tracked manuscript re: distance, island’s info

331-333of the tracked manuscript re: limited human rights in refugees.

I would definitely welcome the eventual publication of these data and insights, as I think the data is so important and MSF’s crucial work on Lesvos warrants significant attention. However, to strengthen the piece, I do suggest a strengthening of the discussion section and close proofread and technical / stylistic edit by a native English speaker.

Thank you, we have attempted to improve the language throughout the 

manuscript.

---

## [Decision Letter · Decision Letter 1]

26 May 2020

PONE-D-19-20179R1

Sexual violence against migrants and asylum seekers. The experience of the MSF clinic on Lesvos Island, Greece.

PLOS ONE

Dear Dr. BELANTERI,

Thank you for submitting your manuscript to PLOS ONE. After careful consideration, we feel that it has merit but does not fully meet PLOS ONE’s publication criteria as it currently stands. Therefore, we invite you to submit a revised version of the manuscript that addresses the points raised during the review process.

We look forward to receiving your revised manuscript.

Kind regards,

Vedat Sar, M.D.

Academic Editor

PLOS ONE

Additional Editor Comments (if provided):

Dear Ms.Belanteri

Thank you for re-submitting your manuscript. Both reviewers see considerable merit in your study, however, one of the reviewers has still concerns you may address in a revision.

We would be glad to see a re-revised version of this manuscript.

Best regards,

Vedat Sar,MD

Reviewers' comments:

Reviewer's Responses to Questions

**Comments to the Author**

1. If the authors have adequately addressed your comments raised in a previous round of review and you feel that this manuscript is now acceptable for publication, you may indicate that here to bypass the “Comments to the Author” section, enter your conflict of interest statement in the “Confidential to Editor” section, and submit your "Accept" recommendation.

Reviewer #2: All comments have been addressed

Reviewer #3: (No Response)

2. Is the manuscript technically sound, and do the data support the conclusions?

Reviewer #2: Yes

Reviewer #3: Partly

3. Has the statistical analysis been performed appropriately and rigorously? 

Reviewer #2: I Don't Know

Reviewer #3: N/A

4. Have the authors made all data underlying the findings in their manuscript fully available?

Reviewer #2: No

Reviewer #3: No

5. Is the manuscript presented in an intelligible fashion and written in standard English?

Reviewer #2: Yes

Reviewer #3: No

6. Review Comments to the Author

Reviewer #2: Thank you for the attentive revision. I am sincerely looking forward to seeing this piece in print. Article needs one more close proofread, however, as several typos remain (for example in lines 111, 217, 246).

Reviewer #3: This topic is certainly timely and the work that the organization has done is admirable. But what this paper presents is less than a scholarly article and reads more like a research report -- and even then a somewhat limited one. There is a descriptive overview of this particular case and context but there is little that is drawn in terms of conclusions or steps for either corrective action, intervention or even suggestions for future study. There were many important findings that were mentioned in passing in the paper but not ever really followed up on. For example, the data suggests significant variance in the gender composition of survivors from what we commonly know, but this is not explored in greater detail. The percentage of survivors who were actually raped is also not commented upon other than briefly. The implication that so-called 'safe' countries are not safe at all was another avenue not followed. These are but a few. I am thus unable to recommend publication even though the topic is significant and it appears that the authors have already gone to some length to address earlier reviews.

7. PLOS authors have the option to publish the peer review history of their article (what does this mean?). If published, this will include your full peer review and any attached files.

Reviewer #2: Yes: Kim Thuy Seelinger

Reviewer #3: No

---

## [Author Response · Author response to Decision Letter 1]

28 Aug 2020

To the editor, PLOS ONE

Dear editor,

Thank you for your message including the reviews of our paper. We have amended the paper following the comments, and we think the paper is better now. In our responses below we refer to line numbers in the new revised version with track changes. Reviewers’ comments are shown, and our response is given point by point in BOLD.

Response to Reviewers

Reviewer #2: Thank you for the attentive revision. I am sincerely looking forward to seeing this piece in print. Article needs one more close proofread, however, as several typos remain (for example in lines 111, 217, 246).

Thank you for your comments. All necessary changes are made accordingly. 

Reviewer #3: This topic is certainly timely and the work that the organization has done is admirable. But what this paper presents is less than a scholarly article and reads more like a research report -- and even then a somewhat limited one. There is a descriptive overview of this particular case and context but there is little that is drawn in terms of conclusions or steps for either corrective action, intervention or even suggestions for future study. There were many important findings that were mentioned in passing in the paper but not ever really followed up on. For example, the data suggests significant variance in the gender composition of survivors from what we commonly know, but this is not explored in greater detail. The percentage of survivors who were actually raped is also not commented upon other than briefly. The implication that so-called 'safe' countries are not safe at all was another avenue not followed. These are but a few. I am thus unable to recommend publication even though the topic is significant and it appears that the authors have already gone to some length to address earlier reviews.

Thank you. 

We recognise the limitations of the research. Thus, this is mentioned in “Discussion”, lines 346-355.

All necessary changes from native English co-author are done to improve the language issues found in the manuscript.

---

## [Editor Report · Decision Letter 2]

2 Sep 2020

Sexual violence against migrants and asylum seekers. The experience of the MSF clinic on Lesvos Island, Greece.

PONE-D-19-20179R2

Dear Dr. BELANTERI,

We’re pleased to inform you that your manuscript has been judged scientifically suitable for publication and will be formally accepted for publication once it meets all outstanding technical requirements.

Kind regards,

Vedat Sar, M.D.

Academic Editor

PLOS ONE

Additional Editor Comments (optional):

The authors addressed reviwers' requests.
---

## [Editor Report · Acceptance letter]

7 Sep 2020

PONE-D-19-20179R2 

Sexual violence against migrants and asylum seekers. The experience of the MSF clinic on Lesvos Island, Greece. 

Dear Dr. Belanteri:

I'm pleased to inform you that your manuscript has been deemed suitable for publication in PLOS ONE. Congratulations! Your manuscript is now with our production department. 

Kind regards, 

on behalf of

Dr. Vedat Sar 

Academic Editor

PLOS ONE